# SKATE : Successive Rank-based Task Assignment for Proactive Online Planning

**Primary Keywords:**
*Multi-Agent Planning;*

## Abstract

The development of online applications for services such as package delivery, crowdsourcing, or taxi dispatching has caught the attention of the research community to the domain of online multi-agent multi-task allocation. In online service applications, tasks (or requests) to be performed arrive over time and need to be dynamically assigned to agents. Such planning problems are challenging because: (i) little or almost no information about future tasks is available for long-term reasoning; (ii) agent number, as well as, task number can be impressively high; and (iii) an efficient solution has to be reached in a limited amount of time. In this paper, we propose SKATE, a successive rank-based task assignment algorithm for online multi-agent planning. SKATE can be seen as a meta-heuristic approach that successively assigns a task to the best-ranked agent until all tasks have been assigned. We assessed the complexity of SKATE and showed it is cubic in the number of agents and tasks. To investigate how multi-agent multi-task assignment algorithms perform under a high number of agents and tasks, we compare three multi-task assignment methods in synthetic and real data benchmark environments: Integer Linear Programming (ILP), Genetic Algorithm (GA), and SKATE. In addition, a proactive approach is nested to all methods to determine near-future available agents (resources) using a receding-horizon. Based on the results obtained, we can argue that the classical ILP offers the better quality solutions when treating a low number of agents and tasks, i.e. low load despite the receding-horizon size, while it struggles to respect the time constraint for high load. SKATE performs better than the other methods in high load conditions, and even better when a variable receding-horizon is used.

## 1 Introduction

In the last couple of years, new applications have emerged where users make requests and a platform has to manage its resources in order to satisfy users' requests. Examples of these application domains are taxi dispatching (Dickerson et al. 2018), ridesharing (Herbawi and Weber 2012), crowdsourcing (Wang, Zhao, and Xu 2020) or package delivery (Cheikhrouhou and Khoufi 2021). All these application domains have led to advances in the field of online multi-agent multi-task assignment. In online service applications, tasks (or requests) arrive over time and need to be dynamically assigned to agents (Dickerson et al. 2018). Such planning problems are challenging because: (i) little or almost no in-

formation about future tasks is available for long-term reasoning; (ii) agent number, as well as, task number can be impressively high; and (iii) an efficient solution has to be reached in a limited amount of time. Regarding the first aspect, the requests arrival model is usually characterized with probability distributions (Mehta 2013) that are then used for planning (Alaei, Hajiaghayi, and Liaghat 2012), (Gong et al. 2022), (Hikima et al. 2022). The issue with this assumption is that beforehand data on the requests is needed to learn the model parameters and it can end up being very specific to a given application rather than general. In the present work, no assumption is made about the arrival requests model.

Another aspect is the nature of resources that can be disposable or reusable. Disposable resources can only be used once, while reusable resources will not be considered as available for new assignments while completing their current tasks but can receive new assignments once they are free (Sumita et al. 2022). The present work assumes reusable resources since this setting is commonly found in many applications, in particular ridesharing platforms. Linked to it, time management for requests assignment is also important. In online settings, planning is triggered for short time intervals. And, at each decision-making step, assignments will consider the requests that have arrived within the last time interval (Alonso-Mora et al. 2017), (Lesmana, Zhang, and Bei 2019). Differently, (Dickerson et al. 2018) assigns requests to agents as soon as they arrive (i.e. continuous planning). Both approaches have qualities and drawbacks: for the first one, and depending on the time interval, it is more likely that (reusable) agents become available allowing for efficient solutions (e.g. path cost) but the waiting time of the requests to be assigned can be important; whereas for the second one, the waiting time decreases as the request is assigned to whatever available agent, however the solution is generally worse (Sumita et al. 2022). Interestingly, (Conforto Nedelmann, Lacan, and Chanel 2023) proposed a proactive approach that can be seen as a compromise between the approaches cited above. It uses a receding-horizon in order to determine which agents will be available in a near-future.

Additionally to the time management aspect, online solving can be particularly challenging because one may design a solving method that has to be fast and efficient (e.g to find efficient solutions in a limited amount of time). It can be

particularly challenging when the number of agents (e.g. resources), as well as, the number of tasks (e.g. requests) is impressively high, a hard combinatorial problem present in several real-life settings.

To address these issues, we propose SKATE, a successive rank-based task assignment algorithm. SKATE is inspired by *MinPos*, a method proposed in (Bautin, Simonin, and Charpillet 2012) and later improved in (Bautin 2013). This meta-heuristic method was developed for the field of multi-robot exploration in unknown environments. *MinPos* assigns (only) a task (e.g. a location to be explored) for each robot. *MinPos* applies a ranking method based on the distance between the tasks and a robot while also taking into account the position of the other robots. In its domain, *MinPos* achieved good performance while being lightweight. SKATE extends the *MinPos* algorithm to address online multi-agent multi-task assignment. SKATE fits the online planning requirements: it assigns several tasks to agents by adapting the metrics and the ranking system, it does not need to assume a task arrival model, and it is able to achieve good solutions even for (very) large problems. Moreover, the proactive approach proposed by (Conforto Nedelmann, Lacan, and Chanel 2023) is nested to SKATE to determine near-future available agents (resources) using a receding-horizon. In this way, SKATE can be seen as a flexible approach that evaluates different sets of (available) agents to define the best assignments.

In the following section, related works are reviewed. Then the planning problem is formally described. Afterwards, the proactive approach and SKATE are presented followed by experiments which results are discussed. Future research directions concludes the paper.

## 2 Related Work

The present work addresses the online multi-task assignment with reusable resources problem. In this class of problem, one may manage the resources (i.e the agents of a fleet) in order to accomplish some tasks efficiently (Khamis, Hussein, and Elmogy 2015), (Hussein and Khamis 2013). In detail, one may minimize the overall traveled distance of the fleet while also minimizing the time between the registration of a request and its execution. This problem is close to the Multiple Traveling Salesmen Problem (MTSP), a classical optimization problem for which several solving methods have been proposed (Cheikhrouhou and Khoufi 2021). According to the taxonomy, our problem is a sub-variant called open-path multiple depots since the agents do not go back to their initial location when they finish the assigned tasks, and the initial position of multiple agents are different.

(Dickerson et al. 2018) introduced the concept of reusable resources, which has been widely used as for example in (Sumita et al. 2022), (Gong et al. 2022) and (Hikima et al. 2022). For time management, both immediate assignment ((Hikima et al. 2022), (Sumita et al. 2022)) and assignment in time intervals ((Alonso-Mora et al. 2017), (Lesmana, Zhang, and Bei 2019)) are common. There are even some works that combine the two aspects such as (Wang and Bei 2022), where the goal is to find a balance between wanting more agents to become available while avoiding the requests withdrawal when the waiting time becomes too long. A majority of works assume a tasks arrival model describing the time of arrival and/or the position of future requests which allows them to reason in long-term. In (Nanda et al. 2020), agents can reject specific requests based on the model. (Hikima et al. 2022) has developed a strategy when the requests arrival model is linked to the rewards for assigning a request to a resource. In (Burns et al. 2012), they determine the likely characteristics of requests a couple of time steps ahead of the assignment then place their agents accordingly. In (De Filippo, Lombardi, and Milano 2019), the authors use previous accounting of requests to build offline a model that gives the more likely scenarios.

Several methods have been used in the literature to solve this optimization problem. The most used method is Linear Programming (LP) and its variants (e.g. Integer Linear Programming). This approach is preferred because it finds the optimal solution (Dickerson et al. 2018), (Sumita et al. 2022). However, it is mainly used for immediate assignment because Linear Programming has a potential high complexity (Basu et al. 2022). In order to respect the online time constraint, it handles only a small number of agents and requests. Genetic Algorithm (GA) is a meta-heuristic approach. The use of GA has been popular since it can provide solutions for large problems in a shorter amount of time than LP. They also provide good performance for multi-criteria objectives (Rangriz, Davoodi, and Saberian 2019). In the context of ridesharing, (Herbawi and Weber 2011) has compared the solution efficiency of a GA and several variants of Ant Colony Optimization. They found that at its best the Ant Colony Optimization gave similar performance than the GA, and that the GA was significantly faster than the Ant Colony Optimization. (Wang, Zhao, and Xu 2020) used a variant of Genetic Algorithm with some elements of Ant Colony Optimization in the context of crowdsourcing. Their approach produced good solutions but was slightly slower than a classical GA.

Finally, in terms of experimental settings, papers focus on a small amount of agents and requests, particularly when using LP. (Sumita et al. 2022) and (Dickerson et al. 2018) apply their methods to a real-life scenario of taxi dispatching in New York, however they limited their evaluation to 30 taxis and 100 or 550 requests, respectively. This is far from reality where hundreds of requests in New York can be received in a time span of 5 minutes. Regarding Genetic Algorithm use, we found that (Herbawi and Weber 2012) has considered 250 drivers and 448 requests for their experiments in the context of ridesharing, which is higher but still much less than what can be present in real-life settings. In the present work, we will show that SKATE can handle much more agents and requests being a promising and competitive approach for online multi-agent multi-task assignment planning problem.

## 3 Problem statement

The multi-task assignment is a combinatorial optimization problem. Over the entire time horizon, we aim to minimize the distance traveled by the fleet of agents (i.e. resources)

while also minimizing the amount of time between the moment a request was registered and the moment it was executed (i.e. task). In the following, we will call the latter criterion the request's waiting time. Generally speaking, we optimize both: user satisfaction while also economizing on the use of resources.

We consider a set of $n$ agents denoted by $A = \{a_1, a_2, \ldots, a_n\}$. The location of the agents is denoted by $\boldsymbol{p_a}$ for all $a \in A$. We assume requests arrive time-to-time throughout the planning horizon $T$. The planning horizon $T$ is divided into small intervals called time steps or time windows. The time windows are indexed by $\tau$ and their duration is a constant equal to $\delta$. For a time step $\tau \in T$, we assume the set of requests $R_\tau$. A request $r \in R_\tau$ can be described by its location $\boldsymbol{p_r}$ and the time it was registered $t_r^b$. Thus, a single request $r \in R_\tau$ can be written as $r = (\boldsymbol{p_r}, t_r^b)$. It is assumed a request can be assigned to only one agent $a \in A$. We note the set of requests that can be assigned to an agent $a \in A$ as $R_a = \{r_a^1, r_a^2, ..., r_a^m\}$, where $m$ is the number of requests assigned to this agent $a$ at a given time. To describe if $r \in R_\tau$ has been assigned to $a \in A$, we use the binary variable $x_{a,r}$ with $x_{a,r} = 1$ and $x_{a,r} = 0$ if that's not the case. The set of requests assigned to $a \in A$ can also be described as $R_a = \{\bigcup_{r \in R_\tau} r | x_{a,r} = 1\}$. Assuming an ordered set $R_a$, the distance between the expected location of agent $a$ and the location of request $r$, and the fact that agents move with a constant speed $v_a$, one can compute the expected execution time of a request $r \in R_a$, noted as $t_r^{ex}$.

Therefore, the general optimization problem we address can be formalized as:

$$\min \left[ \sum_{\tau=0}^{T-1} \left( \alpha \sum_{a \in A} d_{R_a}\left( \bigcup_{r \in R_\tau} r | x_{a,r} = 1 \right) \right. \right.$$
$$\left. \left. + (1-\alpha)\left( \sum_{a \in A} w_{R_a}\left( \bigcup_{r \in R_\tau} r | x_{a,r} = 1 \right) \right) \right) \right] \quad (1)$$

subject to:

$$\sum_{a \in A} x_{a,r} \leq 1, \forall r \in R_\tau \quad (2)$$

with :

$$d_{R_a} = \|\boldsymbol{p_a} - \boldsymbol{p_{r_1}}\| + \sum_{i=1}^{|R_a|-1} \|\boldsymbol{p_{r_i}} - \boldsymbol{p_{r_{i+1}}}\|, \quad (3)$$

$$w_{R_a} = \sum_{r \in R_a} (t_r^{ex} - t_r^b) \quad (4)$$

$\alpha \in [0,1]$ and $(1-\alpha)$ are weights used to balance the importance between the two criteria. The formulation of $d_{R_a}$ was inspired by (Cheikhrouhou and Khoufi 2021) that uses it for a MTSP open-path multi-depot, and denotes the total (Euclidean) path distance from the initial position of $a \in A$ until the position of its last assigned request $r_a^m$. Then, we define the waiting time $w_{r_a}$ as the duration between the time of registration of the request $r \in R_\tau$ and its execution by agent $a$.

The problem with such a formulation is that one may know which requests will be received for each time step $\tau$ in the given horizon $T$. Here, and in several application domains (e.g. taxi dispatching, package delivery, crowdsourcing) little information is available about the requests' arrival, either on their number or their locations. As a result, it is extremely hard to solve this general optimization problem over the entire time horizon $T$.

To treat this problem online with no assumption regarding the requests arrival model, we were inspired by the proactive approach presented in (Conforto Nedelmann, Lacan, and Chanel 2023). In this paper, instead of solving the general optimization problem, they broke it into a small problem at each time step $\tau$ to consider the (new) set of tasks $R_\tau$ with a proactive perspective. For that, they proposed the concept of *available* agents. In classical reactive approaches, *available* agents would be the ones that have already finished executing their previous requests at $\tau$. Whereas in the article cited above, a receding-horizon approach was used for determining the agents to be used at $\tau$: the said *available* agents are the ones that will complete their previously assigned tasks within an immediate horizon $H(k) = k\delta$ with $k \geq 0$ (e.g. $H(5) = 5\delta$ or $H(5) = \tau + 5$). This allows to anticipate the availability of resources and not have to wait for agents to finish their requests before assigning new ones to them. We call $A_\tau(H)$ the available agents at $\tau$ for a certain time horizon $H(k)$.

With this in mind, and adopting a similar proactive approach, we will assign requests at each time step $\tau$ accounting with the agents considered as *available* within the horizon $H$. Thus, we search for a solution for:

$$\min \left[ \alpha \sum_{a \in A_\tau(H)} d_{R_a}\left( \bigcup_{r \in R_\tau} r | x_{a,r} = 1 \right) \right.$$
$$\left. + (1-\alpha)\left( \sum_{a \in A_\tau(H)} w_{R_a}\left( \bigcup_{r \in R_\tau} r | x_{a,r} = 1 \right) \right) \right] \quad (5)$$

subject to:

$$\sum_{a \in A_\tau(H)} x_{a,r} \leq 1, \forall r \in R_\tau \quad (6)$$

## 4  Proactive online task assignment approach

In online service applications, tasks (or requests) to be performed arrive over time and need to be dynamically assigned to agents. In the following, we will first explain the general proactive online process for task assignment.

**General process**  In the general process (illustrated in Alg. 1), at each time step $\tau$, we get the new requests $R_\tau$ that have arrived during the last time window and have been stored in the buffer $B$ (line 6). Then we check the availability at horizon $H$ of the agents (lines 8 - 9). For that we use a variable $t_a^{occ}$ indicating the time at which the agent $a$ will finish its last assigned request. Finally, we use an algorithm to assign the tasks of $R_\tau$ to the available agents $A_\tau(H)$.

Note that this process is general and can implement any task assignment algorithm able to solve the optimization problem presented in Eq. 5. However, this planning problem can be challenging because the number of agents, as well as the number of tasks can be large; yet an efficient solution has to be reached in a limited amount of time. In this context, classical approaches such as Integer Linear Programming, or meta-heuristic approaches (e.g. Genetic Algorithms) may be blocked or only find mediocre solutions. In the following, we propose a method we called Successive Rank-based Task Assignment (SKATE), a simple yet efficient online method for multi-agent multi-task assignment that can be eventually used in line 10 of Algorithm 1. SKATE can be seen as a meta-heuristic solving process.

## 4.1 Successive Rank-based Task Assignment

The main principle of SKATE is the ranking of the requests for each agent while taking into account the expected location of the other agents. We present SKATE in Algorithm 2. SKATE assigns all the registered requests $R_\tau$ to the set of available agents $A_\tau(H)$ of size $n$. This assignment is implemented into several rounds where at each round, a maximum of $n$ requests are assigned. The process is repeated until there is no request left (line 2).

More specifically, to proceed with the assignment we fill a cost matrix $M^C$ of size $|A_\tau(H)| \times |R_\tau|$ where we calculate the cost to assign any request $r_j \in R_\tau$ to any agent $a_i \in A_\tau(H)$ (line 4). Since we are interested in minimizing both the traveled distance and the waiting time, the cost function will be a combination of these two criteria such as:

$$cost(a_i, r_j) = \alpha \frac{d(a_i, r_j)}{v_{a_i}} + (1-\alpha)w_{a_i}(r_j) \qquad (7)$$

with:

$$d(a_i, r_j) = \|\boldsymbol{p_{a_i}} - \boldsymbol{p_{r_j}}\|, \text{ and } w_{a_i}(r_j) = t_{a_i}^{occ} + \frac{d(a_i, r_j)}{v_{a_i}} - t_{r_j}^b$$

where $d(a_i, r_j)$ is the Euclidean distance between the anticipated position of $a_i$ and the position of the request $r_j$. To be able to compare the two criteria, we converted this distance term into time, by dividing it with the velocity of the agent $v_{a_i}$ assumed constant. The waiting time $w_{a_i}(r_j)$ corresponds to the time between the time the request was generated ($t_{r_j}^b$) and the time needed for the agent to go from his expected location to that of the request ($t_r^{ex}$). The term $t_{a_i}^{occ}$

---

Algorithm 1: Proactive online task assignment process

1: Agents $A$ at position $\boldsymbol{p_a}$ and $t_{a_{occ}} = 0, \forall a \in A$
2: Horizon $H = k\delta$
3: $R_{\tau=0} \leftarrow \emptyset$
4: **for** each time step $\tau$ **do**
5:     $A_\tau(H) \leftarrow \emptyset$
6:     $R_\tau \leftarrow \text{GetTasksFrom}(B_\tau, R_{\tau-1})$
7:     **for** $a \in A$ **do**
8:         **if** $t_{a_{occ}} < \tau + H$ **then**
9:             $A_\tau(H) \leftarrow a$
10:     Assign $R_\tau$ tasks to $A_\tau(H)$ agents

---

Algorithm 2: Successive Rank Based Task Assignment

1: **procedure** SKATE($A_\tau(H), R_\tau$)
2:     **while** $R_\tau \neq [\,]$ **do**
3:         $A = A_\tau(H)$
4:         Compute the cost matrix $M^C$ such as $M_{i,j}^C = cost(a_i, r_j), \forall a_i \in A$ and $\forall r_j \in R_\tau$
5:         Compute the rank matrix $M^R$ such as $M_{i,j}^R = Card(\bar{A})$ with $\bar{A} = \{\forall a_k \in A | M_{k,j}^C < M_{i,j}^C\}$
6:         Definition of the variable of ranking S=0
7:         **while** $R_\tau \neq \emptyset$ and $A \neq \emptyset$ **do**
8:             **for** $a \in A$ **do**
9:                 $rank_{min} = min(M_{a,r}^R \forall r \in R_\tau)$
10:                 **if** $rank_{min} = S$ **then**
11:                     assign $r$ to $a$
12:                     $t_r^{ex} = w_a(r)$ and $t_a^{occ} = t_a^{occ} + \frac{d(a,r)}{v_a}$
13:                     $A = A \setminus a$ and $R_\tau = R_\tau \setminus r$
14:             S = S+1
15:         **for** $a \in A$ **do**
16:             Get last request $r$ assigned to $a$
17:             $\boldsymbol{p_a} = \boldsymbol{p_r}$
18: **end procedure**

---

refers to the time the agent is considered busy (e.g. executing previous requests) until it can execute $r_j$.

To rank the requests for an agent $a_i \in A$, we will not simply consider only the value of the cost for that agent, but also take into account the costs of the other agents for each specific request $r \in R_\tau$ (line 5). For that, we look at the cost $M_{k,j}^C$ for $A = A \setminus a$. If all $M_{k,j}^C > M_{i,j}^C$, then it means $a_i$ is the most interesting agent to assign that request to and get $M_{i,j}^R = 0$. Otherwise, its ranking is equal to the number of agents that have a lower cost for that request. Note, some agents can have multiple requests ranked 0 (they have lower costs for more than one request) and some can have none. If two requests have the same minimal ranking value, then we assign to the agent the request with the lower cost. The assigned request and the agent then both become occupied (line 13) and we update the waiting time of the assigned request and the time until which the agent is occupied (line 12). When all of the agents have become occupied, we consider that this round of assignment is over and start the process again (line 3). For the new round, we update the position of the agents using the position of the last request assigned to each of them (line 17).

**Complexity of SKATE** As we can see in Algorithm 2, the two necessary inputs are the available agents and the requests to assign. To analyze the complexity of SKATE, we consider that there are $n$ agents and $m$ requests. First consider the case where the loop of line 2 is only done once. The filling of the cost matrix in line 4 has a cost of $\boldsymbol{mn}$. And, ranking these requests for each agent while taking into account the position of all of the other agents is $\boldsymbol{n^2 m}$ expensive. We now go back to the original algorithm, going through the loop of line 2 several times, which means that

several requests have to be assigned at least to an agent. After the first assignment round, there are $m - n$ requests left. This means that to assign all the requests, the process of filling the cost matrix, ranking matrix, and assignment must be done $\frac{m}{n} - 1$ times. In the following, we detail the calculation of the complexity $C(m, n)$ or $C$ of SKATE for the the $\frac{m}{n} - 1$ rounds:

$$
\begin{aligned}
C = &\overbrace{\sum_{i=0}^{\frac{m}{n}-1} n(m-in)}^{\text{Cost matrix filling}} + \overbrace{\sum_{i=0}^{\frac{m}{n}-1} n^2(m-in)}^{\text{Ranking}} + \overbrace{\sum_{i=0}^{\frac{m}{n}-1} n^2(m-in)}^{\text{Assignment}} \\
= &\sum_{i=0}^{\frac{m}{n}-1} mn - n^2 \sum_{i=0}^{\frac{m}{n}-1} i + 2\left(\sum_{i=0}^{\frac{m}{n}-1} mn^2 - n^3 \sum_{i=0}^{\frac{m}{n}-1} i\right) + \\
&+ \frac{m}{n}mn - n^2 \sum_{i=0}^{\frac{m}{n}-1} i + 2\left(\frac{m}{n}n^2 m - n^3 \frac{m}{2n}\left(\frac{m}{n} - 1\right)\right) \\
= &\, m^2 - \frac{1}{2}n^2 \frac{m}{n}\left(\frac{m}{n} - 1\right) + 2\left(m^2 - n\frac{m^2}{2} + \frac{n^2 m}{2}\right) \\
= &\, m^2 + \frac{mn}{2} + m^2 n + n^2 m = \boldsymbol{m^2 n + n^2 m} \qquad (8)
\end{aligned}
$$

In the next section, we present the methodology applied to evaluate the proactive online multi-agent multi-task assignment approach using SKATE.

# 5 Experiments & Results

To evaluate the impact of the use of the proactive approach, we first compute the assignments based on a reactive approach (no anticipation about agents availability), that we note $H(0)$, then using the receding-horizon to anticipate resources from one to five time windows, denoted as $H(1)$ to $H(5)$. We also use a variable receding-horizon $H(v)$ as proposed by (Conforto Nedelmann, Lacan, and Chanel 2023), where the assignments using different receding-horizons (from $H(0)$ to $H(5)$) are computed in parallel and the best solution is chosen. To evaluate the solutions proposed by SKATE, assignments are also computed using two baselines from the literature. More specifically, a baseline applying Integer Linear Programming (i.e. branch-and-bound algorithm), and another one applying a Genetic Algorithm. Finally, assignment methods are evaluated on two benchmarks mainly to analyze their solution efficiency and scalability. The first benchmark is a synthetic one for which we have a constant number of requests arriving every $\tau \in T$. The second benchmark, already used in the literature (Sumita et al. 2022), is based on registered requests of taxis in New York in January 2013 [1]. This realistic scenario is particularly interesting for mimicking a high-load problem, where we need to find, online, assignments for a high number of requests and agents in a limited amount of time. In all experiments and for all methods the $\alpha$ parameter was set to $0.75$, putting more evidence on the traveled distance criterion.

**Metrics** The metrics used to quantify the impact of the use of a receding-horizon and to compare the efficiency of the

---

[1]available at: http://www.andresmh.com/nyctaxitrips/

three solving methods are: (i) the overall distance traveled by the (fleet of) agents; (ii) the percentage of assigned requests; (iii) the average waiting time for requests; and (iv) the time necessary for computing the assignments.

## 5.1 Baselines

**Integer Linear Programming** To build the Integer Linear Programming (ILP) model we were inspired by the work presented in (Kara and Bektas 2006). This work details the objective function and constraints of the classical MTSP and variants, including the MTSP open-path multi-depot. In order to assign a sequence of requests $R_a$ to an agent $a \in A$, we use the set $L = A_\tau(H) \cup R_\tau$ and a the cost matrix $M^C$ of size $|L| \times |L|$ where $M_{i,j}^C = c_{i,j}, \forall(i, j) \in L^2, i \neq j$. Each element on this matrix refers to the cost between agents and requests (including between 2 agents or 2 requests), such as:

$$
c_{i,j} = \alpha \frac{d(i,j)}{v_a} + (1 - \alpha)w_a(i, j), \ \forall i, j \in L
$$

The objective function and its constraints are defined as:

$$
\text{minimize} \left( \sum_{i \in L} \sum_{j \in L} c_{i,j} x_{i,j} \right) \qquad (9)
$$

subject to:

$$
\sum_{i \in L} \sum_{j \in A_\tau(H)} x_{i,j} = 0 \text{ and } \sum_{j \in R_\tau} x_{i,j} \leq 1, \ \forall i \in L
$$

$$
\sum_{i \in L} x_{i,j} = 1, \ \forall j \in R_\tau \text{ and } x_{i,j} + x_{j,i} \leq 1, \ \forall(i, j) \in L^2
$$

The first constraint clarifies that we can not assign an agent to another agent or a request to an agent since an agent is supposed to move towards requests. The second models the last request an agent has to complete (open-path condition in the MTSP). The third imposes that all the requests must be assigned. The last constraint avoids an agent going back and forth between two positions. To solve this model, we use the Gurobi Optimizer, which employs the branch-and-bound algorithm to find a solution for the ILP problem. The solver explores possible solutions and tries to find the optimal one. Note that the branch-and-bound can be time-consuming: in the worst-case scenario we have a complexity of $O(2^n)$ (Basu et al. 2022), which is significantly more than SKATE ($O(m^2 n + n^2 m)$).

**Genetic Algorithm** The Genetic Algorithm (GA) is an evolutionary algorithm inspired by the natural selection process. The general principle is that we start with a random set of chromosomes called population (e.g. initial assignments). These chromosomes will be evaluated and the ones with the better scores (e.g. cost) will be used to create a new population (e.g. new set of solutions). This way, the population can hopefully improve generation after generation (regarding their scores). The operations used to build a new generation are the crossover (where we take two parent chromosomes and fuse them) and mutations (where we swap the order of the requests or swap two requests between two different agents). The algorithm stops when a stopping condition is reached, in general, either the solution is no longer

improving or a time limit is reached. For the problem addressed in this work, each chromosome represents a specific assignment of the requests to the available agents. We have almost used the same approach structure and parameters as (Conforto Nedelmann, Lacan, and Chanel 2023). However, the optimization problem we tackle differs. As we focus on minimizing both the overall traveled distance of the agents and the waiting time of the request, the fitness function we use is different and is expressed as follows:

$$\alpha \Big( \sum_{a \in A_\tau(H)} d_a \frac{1}{D_{max}} \Big) + (1-\alpha) \Big( \sum_{a \in A_\tau(H)} w_a \frac{1}{W_{max}} \Big)$$

where $d_a$ is the distance traveled by $a \in A$ to execute all its $R_a$ requests in the proposed chromosome, $w_a = \sum_{r \in R_a} w_r$ as the sum of the waiting time of the requests assigned to $a$ in this chromosome. To compare it to the first generation, we normalize each term of the fitness by $D_{max}$ and $W_{max}$, which are the maximum values obtained in the first generation for the total traveled distance and waiting time respectively. We consider these values as the worst case because the first generation is filled randomly.

## 5.2  Synthetic Benchmark Experiments & Results

**Setup**  We consider two different set-ups: for the first, at each time step 20 requests are randomly placed in the workspace defined by a square of 10m x 10m; the second with 50 requests being generated at each time step. For both of them, the fleet will be composed of 10 agents which move at a constant speed of $1m/s$. The duration of a time window is 5 seconds (i.e. time step interval). The total simulation horizon is composed of 30 time steps. The simulation process is executed 10 times. In this synthetic benchmark, we consider that the requests are executed when the agents reach their locations.

**20 requests every time step**  The results in this set-up are illustrated in Figure 1. Regarding the percentage of assigned requests (Figure 1b) almost all of them are assigned for all three methods, confirming they all can handle such a scenario. In terms of traveled distance (Figure 1a), for all receding-horizons, the ILP method gives the best results. This is expected since the ILP looks for (sub-)optimal solutions. Between the GA and SKATE, the GA is more interesting when using the reactive approach ($H(0)$), however SKATE produces better values when using the proactive approach ($H(k)$ with $k > 0$). Note that the proactive approach provides improvements for all solving methods helping to reduce the traveled distance for the ILP, with a most important impact on these values for the GA and SKATE methods. For all three methods, the better results are given when using the variable receding-horizon $H(v)$ and in that case, SKATE achieves values close to the ILP ones. Table 1 shows the mean waiting time for the requests for ILP, GA and SKATE for $H(v)$. Once again, ILP gives the better solution however SKATE solutions are competitive. GA has a larger mean than the others, which is coherent with its higher traveled distance. Figure 1c compares the solving time of the three methods, i.e. the time necessary for computing assignments. Central line dots correspond to the mean time for a given

| | Assignment method | | |
|---|---|---|---|
| | ILP | GA | SKATE |
| $R_\tau = 20$ | 22.52 | 33.6 | 23.5 |
| $R_\tau = 50$ | 51.03 | 263.96 | 55.87 |

Table 1: Average waiting time of assigned requests (in seconds) for $A = 10$ and $H(v)$.

time step calculations across the 10 executions. The shadow area corresponds to the min-max values. We notice the ILP method is already reaching its time limit at 5 seconds, which corresponds to the time limit for online computations and the duration of the time window $\delta$. The GA method needs, on average, 3 seconds and SKATE less than a second. This highlights that SKATE is much more light-weighted than the other methods used and suggests it could be particularly interesting in higher-load setup. As a result, ILP gives the best performances (followed by SKATE and then GA) but is almost reaching the time budget in this set-up.

**50 requests every time step**  The results for this heavier set-up are given in Figure 2. The first observation is about the disparity in terms of the percentage of assigned requests shown in Figure 2b. SKATE manages to assign close to 100% of the requests in general, though we notice a slight improvement using the proactive approach compared to the reactive approach. For GA, the percentage of the assigned requests is lower than for SKATE but it still manages to assign almost 90% or more of the requests both for reactive and proactive approaches. In particular, this improvement is even more noticeable when the size of the receding horizon increases, and for $H(v)$, the value achieved is close to the SKATE one. Surprisingly, the outlier results concern ILP which is far from assigning all the requests. We investigated why these percentages are so low and we discovered that if agents are unavailable two subsequent time steps (which means there would be 150 requests to assign at the next time step), the ILP approach can not handle such a load and reaches the time limit before managing to find any solution. In the same situation, GA and SKATE are able to handle this additional load. We notice improvements in solution efficiency for ILP when the proactive approach is used: anticipating the availability of the agents allows to delay or even not encounter the problem of getting stuck. These results are detailed in supplementary materials. In terms of traveled distance (see Fig. 2a), SKATE gives better results than the GA regardless of the size of the receding-horizon. The ILP method shows fluctuations but we speculate it is linked to the low number of assigned requests. As the size of the receding-horizon grows, the traveled distance for the ILP also grows since more requests have been assigned. But this does not give us the optimal: when a solution is obtained, it is in majority a sub-optimal solution. Regarding the average request waiting time (see Table 1), ILP still gives the lowest value. However, it only gives partial information since this average only takes into account the assigned requests. SKATE has an average waiting time which is 4 seconds higher than ILP however it assigns more requests than ILP. As a result, we judge that ILP is not able to handle a higher-load set-up since it could not produce relevant so-

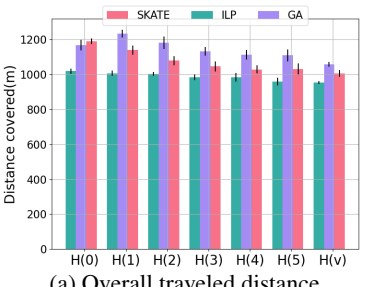
(a) Overall traveled distance

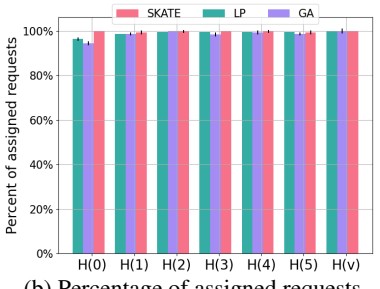
(b) Percentage of assigned requests

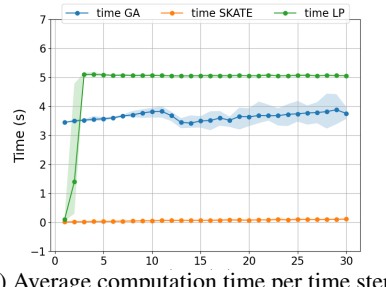
(c) Average computation time per time step.

Figure 1: Comparison of reactive and proactive approaches for ILP, GA and SKATE and for $R_\tau = 20$.

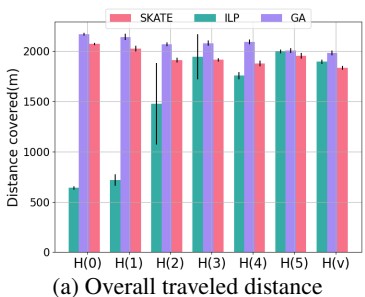
(a) Overall traveled distance

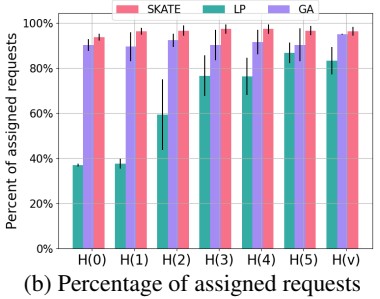
(b) Percentage of assigned requests

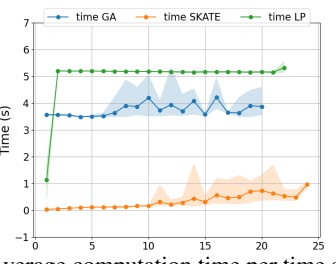
(c) Average computation time per time step.

Figure 2: Comparison of reactive and proactive approaches for ILP, GA and SKATE for $R_\tau = 50$.

lutions in the defined computation time limit. Between the GA and SKATE, SKATE gives consistently better results whether in terms of distance, number of assigned requests or average waiting time.

### 5.3 Real-life data Experiments & Results

**Setup** To confront solving methods to realistic settings, we exploit an open data set listing taxi requests in New York City. In this data set, a request characterizes a real-life taxi call with a starting point located at $p_{r_s}$, a final point at $p_{r_f}$, and a request registration time. In this benchmark, we consider the pickup time as the request registration time $t_{r_b}$, and the request is executed once the agent has reached the destination location. The distance cost is then adapted to account as: $d_{a,r} = \|p_a - p_{r_s}\| + \|p_{r_s} - p_{r_f}\|$. This additional distance is also taken into account for the waiting time calculation. The total distance for an agent to execute all its assigned requests is defined as:

$$d_a = \|p_a - p_{r_{1_s}}\| + \sum_{i=1}^{m-1} \|p_{r_{i_s}} - p_{r_{i_f}}\| + \sum_{i=1}^{m-1} \|p_{r_{i_f}} - p_{r_{i+1_s}}\|$$

We evaluate SKATE and GA methods with a reactive and proactive approach over 3 nights, from January 7th to January 9th 2013 from 12AM to 7AM. Each night corresponds to an independent simulation. The time window $\delta$ (i.e. difference between two time steps) has a duration of 5 minutes. We consider a fleet of 1000 agents traveling at a constant velocity of 30 mph (the speed limit in New York). We included the representation of the varying number of requests arrival in supplementary materials. Contrary to the previous benchmark, here we address a variable number of requests arriving throughout the simulation. Moreover, such a simulation has a heavier load than the previous benchmark, since we

have more requests and agents (1000 agents and 200-1200 requests) and still a limited amount of time for calculations. Since ILP was already struggling in the previous benchmark, we do not include it in this part. Instead, we will be focusing only on the GA and SKATE results.

**Comparison for 1000 agents** The results of experiments using real-life data are presented in Figure 3. Looking at the percentage of assigned requests in Fig. 3a, SKATE is more efficient than GA. However, SKATE is not able to assign more than 80% of the requests. The reason is the significant amount of requests arriving between 6 and 7 AM. With that amount of requests (end of the simulation), the agents stay busy for more time than the receding-horizon size, being not available until the end of the simulation. GA is further impacted because of less efficient assignments: the agents cover a greater distance being busy for an even longer time. In the time limit of 5 minutes, GA only computes a couple of generations, preventing GA to improve solutions. It is confirmed by the solving time (see Fig. 3c). GA struggles to find an efficient assignment within time budget when compared to SKATE. Poor solutions also have an impact on the waiting time which is bigger for GA than for SKATE (results detailed in supplementary materials). Interestingly, the proactive approach increases the percentage of assigned requests for both approaches but the progression is more important for GA. Additionally, we compare the time needed for assignment calculations and the theoretical complexity for SKATE shown in Figure 5 for $H(v)$. The similarity of curves shows that the complexity we calculated is correct.

**SKATE with different fleet sizes** The experiments above showed that a fleet of 1000 agents is not sufficient to handle all of the requests. To estimate how many agents would be necessary, we studied the impact of the fleet size on the per-

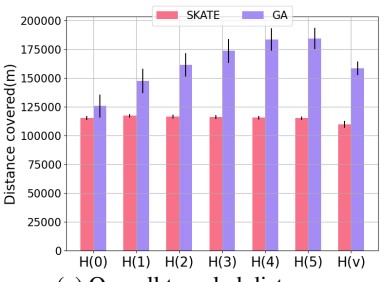
(a) Overall traveled distance.

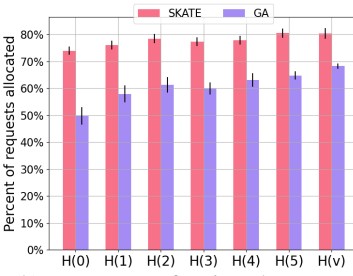
(b) Percentage of assigned requests.

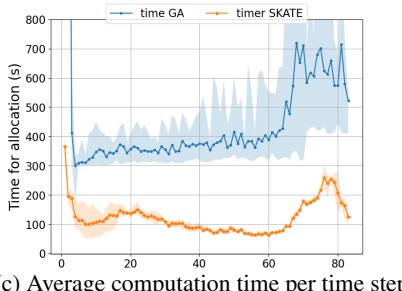
(c) Average computation time per time step.

Figure 3: Comparison of reactive and proactive approaches for GA and SKATE in real-life settings.

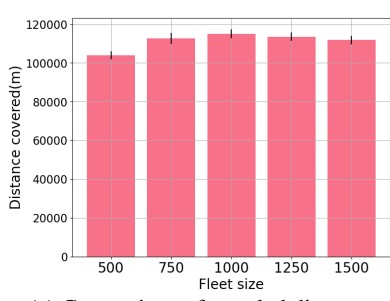
(a) Comparison of traveled distance.

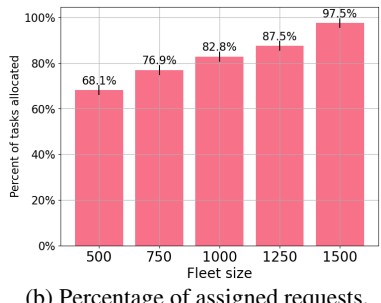
(b) Percentage of assigned requests.

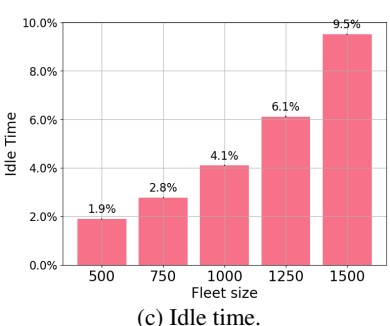
(c) Idle time.

Figure 4: Comparison of the overall distance, number of assigned requests and idle time regarding the fleet size.

centage of assigned requests and waiting time.To see how occupied the agents are, we have recorded how much time the agents stay idle waiting for a new assignment. We tested a fleet of 500, 750, 1000, 1250 and 1500 agents.

Results are compiled in Figure 4 and Table 2. They summarized only the results for the variable receding-horizon $H(v)$ since it consistently gives the best results in previous experiments. First, Fig. 4a shows only a small variation of the overall distance covered by agents with SKATE. We notice that considering there are more agents, the individual distance traveled by agents is lower. It also suggests a better distribution of the requests across the agents, as we see in Fig. 4b that the number of assigned requests increases with the number of agents and gets close to 100%. In terms of waiting time (see Tab. 2), it decreases when the number of agents increases since the better distribution leads to less waiting time. We note a sort of threshold around 15 minutes, waiting time decreases marginally for more than 1000 agents. We speculate it is linked to the physical aspects of the problem (distance between the starting and final point). Looking at idle time (i.e. when agents wait for their new assignments) in Fig. 4c, it increases alongside the increase of

| | Size of the fleet | | | | |
|---|---|---|---|---|---|
| | 500 | 750 | 1000 | 1250 | 1500 |
| $\bar{w_r}$ | 51.52 | 29.44 | 16.3 | 14.64 | 13.36 |

Table 2: Average waiting time $\bar{w_r}$ (in minutes).

the number of agents. It is mainly due to the agents having to wait one or several rounds for their first assignment at the start of the night, and a smaller part is due to the lower amount of requests between 3:30AM and 5:00AM.

# 6 Conclusion and Future work

One of the main challenges of the online multi-agent multi-task assignment field is to design solving methods able to find efficient solutions while scaling well (lightweight) given the combinatorial aspect of such planning problems. We proposed SKATE, a successive rank-based task assignment algorithm for online multi-agent planning. Built upon a ranking method considering agent-to-task costs, SKATE successively assigns a task to the best-ranked agent until all tasks have been assigned. We compared SKATE with two baseline methods already used for online planning: GA and ILP. We found from a theoretical and experimental approach that SKATE is faster than the other methods. For light-load, ILP gives better cost-wise results, however in high-load, ILP struggles whereas SKATE scales well giving efficient solutions. SKATE and GA were also compared in real-life settings (e.g. taxi dispatching). SKATE is the best method between the two. Regarding the results achieved with SKATE and different fleet sizes, we plan to further improve our approach to also determine, online, the correct fleet size to met current demands. For instance, by reducing the number of agents when requests are low or by increasing it when the number of requests increases.

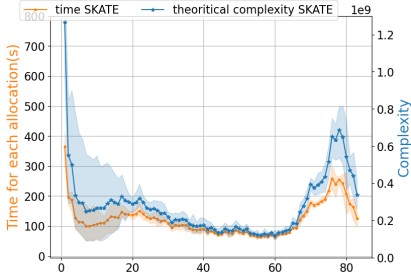

Figure 5: Time and complexity for SKATE per time step.

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
