# OpenReview forum: "SKATE : Successive Rank-based Task Assignment for Proactive Online Planning"
_icaps-conference.org/ICAPS/2024/Conference — ICAPS 2024_

### Official Review · Reviewer_Vim4 · 2024-01-03

**Significance And Importance:** 1
**Soundness:** 2
**Novelty:** 2
**Clarity:** 2
**Overall Evaluation:** 1
**Confidence:** 4

**Weaknesses:**

-1: Major weaknesses requiring significant work to be addressed for the paper to be accepted.

**Contributions Of The Paper:**

The paper presents a greedy iterative approach to assign tasks to agents. At some time intervals, the available (either now or in the short future) agents are iterated over to be assigned the current set of requests (tasks). The best agent is assigned at each iteration until there are no more requests to be assigned.

**Ethical Considerations:**

(1) Not Applicable: The paper does not have any ethical considerations to address

**Nomination For Best Paper:**

No

**Questions For Authors:**

Why did you define the ILP in pg 5 by mixing agents and resources in the cost matrix?

**Reproducibility:**

3: Authors describe the implementation and domains in sufficient detail.

**Strengths Of The Paper:**

- The paper addresses a real-world task, that of online task assignment.
- Experiments in different sizes of tasks, including hard tasks.
- The approach scales to a high number of agents and tasks.
- The paper is relatively easy to follow.

**Weaknesses Of The Paper:**

- The paper contains "online planning" in the title, but there is no planning involved. The task the authors are trying to solve is an online (dynamic) assignment problem, where there are agents and tasks, and a cost associated to assigning an agent to a task. However, once the tasks have been assigned, there is no planning involved. Tasks are solved by just one action.

- I would suggest including a more straightforward algorithm as a baseline for comparison. Random assignment would be one, but very weak. But, you can easily define a greedier approach to yours where you would assign to each new request the least costly agent with complexity O(nm).

- I do not see a reason to model the ILP in pg 5 as you have done by mixing agents and requests together as rows/columns of the matrix. I think it makes the approach highly inefficient by making the matrix unnecessarily large. This could explain, only partially, the poor results when scaling up.

- Notation needs some improvement. As examples:
  * in line 13, T is defined as a time horizon, and then in line 216 it is used as a set (\tau\in T).
  * you use \tau as an index for time windows and as a time step
  * the definition of r only contemplates one location, but in many domains (as you also acknowledge in the experiments) a request might need two locations (the place to pick up some good/agent and the place to drop that good/agent).
  * the definition of R_a in line 227 includes the time stamp \tau. So, it should refer to it and it is not the "set of requests assigned to a", but "set of requests assigned to a at \tau". And R_a_\tau
  * in Equation 1, I do not understand the notation of the objective function. You compute the product of d_R_a which is a number and an union of requests, which is a set. The same applies to w_R_a
  * the notation of the MILP would also require the definition of x_a,r to be 0 or 1.
  * line 241, w_r_a -> w_R_a
  * line 6 of Algorithm 1, what is B_\tau -> R_\tau ?
  * is line 10 a call to Algorithm 2? Say it explicitly if so

Minor issues:
- math mode for references to S in Algorithm 2
- the the (line 364)
- a the (line 409)

AFTER REBUTTAL
I thank the authors for the rebuttal. It has helped me move up my recommendation. I would strongly suggest you include the topics of the discussion in the final version of the paper if accepted. In relation to the discussion on planning, I was referring to the fact that the tasks are single-step actions. It was not clear to me reading the paper that agents could be assigned a sequence of tasks in the same reasoning step. Given that, it is clear now why you created such a matrix. Probably, you could stress that in the final version. I would also suggest including a modified discussion wrt to the state of the art on related tasks as the ones suggested by other reviewers on VRP and related tasks.

---

> ### Author Rebuttal · Authors · 2024-01-26
>
> Thanks for the time spent reading and reviewing our paper. We will consider the suggested corrections.
>
> Concerning the title of the article, in our view, and as defined by Nau,D., Ghallab, M. and Traverso, P. (2004), planning is the process of choosing and organizing actions while anticipating their effects. At each time step, SKATE defines a sequence of tasks (i.e. actions) that each agent must achieve. Thus, we considered that our work can be seen as planning. We understand that since we did not provide this definition, it was unclear and we apologize for the confusion.
>
> The authors of MinPos, which SKATE is based, have already shown the advantages of the ranking step when compared to a greedy strategy (see Fig.1\&5 in reference Bautin et al. 2012). However, differently from MinPos, SKATE assigns a sequence of tasks to each agent. Our simulations (not reported in the paper) confirmed their results: the greedy method achieved 15% higher average distances and 25% higher average waiting time than SKATE. Thus, we did not include this comparison given its partial novelty. As SKATE is based on MinPos, it preserves the same properties regarding the distribution of tasks between agents and conflicts resolution.  And in our opinion, the comparison with ILP and GA is a novel, fair and a more appropriated comparison. Moreover, it shows that SKATE achieves good cost-wise solutions.
>
> Concerning the ILP matrix, we defined it like this for two reasons. Firstly, we followed the formulation proposed in reference (Kara \& Bektas 2006). We replicated their objective formulation and constraints since the cited article and ours address an open-path multiple depots variant of MTSP (lines 131-136). Secondly, as we needed to assign a sequence of requests to each agent, we need both the agent-to-task and the task-to-task costs to determine the cost of a given sequence, and then chose the sequences that minimize the objective function for all agents. One may argue that we could implement successive ILP solving, assigning only one task to each agent per round and using only agent-to-task cost in the cost matrix. But this would mean that the solution regarding the set $R_\tau$ of tasks could be not the optimal one, which was our target.
>
> We hope our answers will clarify all misunderstandings and convince you about the relevance of our work.
>
> Nau,D., Ghallab, M. and Traverso, P. (2004). Automated Planning: Theory and Practice

---

### Official Review · Reviewer_y9dj · 2024-01-22

**Significance And Importance:** 2
**Soundness:** 4
**Novelty:** 3
**Clarity:** 4
**Overall Evaluation:** 2
**Confidence:** 5

**Weaknesses:**

1: Minor weaknesses that are easily fixable.

**Contributions Of The Paper:**

This paper presents a successive rank-based task assignment algorithm for online multi-agent planning named SKATE. The core idea of SKATE is to successfully assign tasks to the best-ranked agent using a meta-heuristic approach inspired by another method, MinPos. SKATE extends MinPos by allowing multiple tasks to be assigned to an agent using a ranking system. Another major feature of SKATE is that there is no need for information on task arrival distributions, which is crucial for some other existing models. Both the theoretical complexity and experimental results are conducted and well discussed. Compared to existing models such as Integer Linear Programming~(ILP) and Genetic Algorithm~(GA), SKATE manages to achieve better performance in high-load conditions and the performance can be further improved by combining variable receding horizons. SKATE is also considered ``lightweight'' in terms of its ability to scale to different sizes of problem settings.

**Ethical Considerations:**

(1) Not Applicable: The paper does not have any ethical considerations to address

**Nomination For Best Paper:**

No

**Questions For Authors:**

1. What criteria are used to determine the value of $\alpha$? Isn't it more reasonable to use a different value of $\alpha$? For instance, when more requests are coming, the system might be more likely to suffer delays. Consequently, isn't that the wait time should be considered more over distance traveled?

2. For the real-life data, are you still using the Euclidean Distance between different locations? Isn't it supposed to be the actual feasible path to the goal? For instance, two locations might be close in Euclidean Distance but detours are needed since the roads are not well-connected. Or maybe these scenarios are rare since the New York dataset is well-connected everywhere? Can you further clarify this?

**Reproducibility:**

3: Authors describe the implementation and domains in sufficient detail.

**Strengths Of The Paper:**

This paper is well-written, smooth, and technically sound. The author did a great job explaining the current challenges in task assignments for online multi-agent planning scenarios. The rationale and purpose behind the design of SKATE are clearly and effectively presented and discussed.

The Experiment section is also well formulated. The other methods such as ILP and GA, which SKATE is comparing against are also presented in enough detail. Both the synthetic and real-life data are being tested and discussed in sufficient detail.

**Weaknesses Of The Paper:**

I don't see any major weaknesses in this paper. Please refer to the rebuttal questions for more detailed comments.

Minor issues:
1. I didn't find any description of the rank matrix in the Algorithm.2, $M^R_{i, j}$, $Card(\bar{A})$. Adding one or two more sentences addressing it would be nice.

---

> ### Author Rebuttal · Authors · 2024-01-27
>
> We thank you for your informed opinion, insightful feedbacks, and for highlighting the importance of our work.
>
> Firstly, we address the issue concerning the ranking matrix  $M^R$ and we apologize for that. The $M^R$ matrix uses the cost matrix  $M^C$, which details the cost for assigning each agent $a_i$ to all the requests $r_j$, in the following way: to build each value $M^R_{i,j}$, each agent $a_i$ will determine how many other agents $a_k$ have a lower cost for that request $r_j$. If no other agents have a lower cost, then $a_i$ has the lower ranking with $M^R_{i,j}=0$. Otherwise, the ranking is equal to the number of agents with a lower cost $M^R_{i,j}= |a_k|$ if $M^C_{k,j}<M^C_{i,j}$.
>
> We now address your questions. Regarding the $\alpha$ value, we carried out empirical tests for values of 0, 0.25, 0.5, 0.75, and 1. The value of 0.75 provided the best balance between the distance and the waiting time criteria for all methods, and was then kept for experiments (lines 393-395 in the paper). We will clarify this point in the final version of the paper.  Varying the $\alpha$ value in function of requests load is an insightful idea that we will certainly implement in future work. Indeed, when the requests load is higher, it would be more advisable to prioritize the waiting time criterion over the distance criterion. Thank you.
>
> Concerning your second question, we used the Euclidean distance between the different locations. While it lacks a bit of realism, we made this choice for a couple of reasons. SKATE itself is not called into question when using this metric across the different methods (e.g. ILP, GA).  Future work could incorporate more realistic path costs. The other reason is that this metric can be a valuable metric for applications using aircraft such as drones or even flying taxis, which can fly in a line-of-sight way for instance.
>
> We once again want to thank you for suggesting ideas and for your recommendation.

---

### Official Review · Reviewer_XQMJ · 2024-01-23

**Significance And Importance:** 2
**Soundness:** 3
**Novelty:** 2
**Clarity:** 4
**Overall Evaluation:** 1
**Confidence:** 3

**Weaknesses:**

0: Minor weaknesses requiring some work to be addressed for the paper to be accepted.

**Contributions Of The Paper:**

The paper describes an online task assignment problem, where a team of agents are required to accomplish continuously generated tasks within the operation period.

The proposed method assigns incoming tasks at each timestep for agents to finish their current tasks in near time windows and minimises both travel distance and waiting time. It ranks the potential assignments using how many assignments are better and the travel cost.

The experiment shows the proposed approach is fast and looking ahead to agents going to finish tasks in the future improves the solution quality for the proposed method.

**Ethical Considerations:**

(1) Not Applicable: The paper does not have any ethical considerations to address

**Nomination For Best Paper:**

No

**Questions For Authors:**

How is the ranking method different from ranking on cost? any live example to tell the difference?

Any results on raking just on travel cost?

In the Real-life data Experiments, does 200 - 1200 requests indicate 200 - 1200 requests per timestep?

Can agents be assigned multiple tasks (a sequence of tasks) at the same time?

**Reproducibility:**

3: Authors describe the implementation and domains in sufficient detail.

**Strengths Of The Paper:**

The paper is written mostly clear and easy to follow.

The proposed method is simple and demonstrated its efficiency advantage in the experiments.

**Weaknesses Of The Paper:**

The literature review lacks discussion on online VRP or online TSP studies.

The novelty is limited. Online VRP or TSP is not new, and the proposed algorithm simply assigns tasks with relatively small travel costs. It is not clear how the additional ranking step makes the assignment different from just ranking with travel cost, as the experiment didn't evaluate the effectiveness of the ranking method.

Looking at whether agents will finish their tasks in the future time window is a very simple concept, and many other works consider agents to be assigned a sequence of tasks, which means they will append tasks before agents finish all their tasks as well.

The algorithm evaluated given fixed tasks release data, how fleet size influences results. But didn't evaluate how the performance is impacted by the increasing of tasks incoming each timestep.

---

> ### Author Rebuttal · Authors · 2024-01-27
>
> Thanks for the time spent reading and reviewing our paper.
>
> *Can agents be assigned multiple tasks at the same time?* At each time step SKATE assigns a sequence of tasks to each agent via successive assignments(lines 308-310): after a first ranking process, it assigns one task per agent, then projects agents to their future locations and repeats the ranking and assignment processes until no task is left.
>
> *In Real-life data, do 200-1200 requests mean 200-1200 requests per timestep?* Yes. We checked how an increasing number of tasks impacted results since in this scenario, requests go from 200 to1200 between 5AM-7AM. SKATE was able to handle this load with several fleet sizes(from 500 to 1500 agents).
>
> *How is the ranking method different from ranking on cost?Any example to tell the difference?* Assigning tasks based only on cost(greedy method)results in a local optimization per agent whereas SKATE ensures a better distribution of tasks between agents. The authors of MinPos, which SKATE is based on, showed the advantage of ranking compared to a greedy method, see Fig.1\&5 in Bautin et al(2012). Our simulation(not reported)confirmed their results: the greedy method had a 15% higher average distance and 25% higher average waiting time than SKATE.
>
> *Any results on ranking just on travel cost?* SKATE uses $\alpha$ to balance the travel cost and the waiting time criteria. Note $\alpha=1$ corresponds to considering only the travel cost in ranking. After some empirical tests, we found that 0.75 gave the best balance for all methods, and was then used for experiments(lines 393-395).
>
> *Online VRP & TSP*: there are online TSP works such as Filippo et al(2019) or Bampis et al(2022), both using a priori information of requests to build a distribution of the more likely scenarios or to plan trajectories whereas we do not use a priori information. Patel et al(2020) addressed online VRP for ridesharing where agents have a fixed goal and must make detours. Our problem is different: open-path multi-depot MTSP(lines 133-136). Due to lack of space, we cut out these papers from the related work part.
> We hope our answers clarify misunderstandings and convince you about the relevance of our work.
>
> Filippo et al(2019). How to tame your anticipatory algorithm, IJCAI
>
> Bampis et al(2022). Online TSP with Known Locations, Algorithms and Data Structures
>
> Patel et al(2020). Hybrid Genetic Algorithm for Ridesharing with Timing Constraints: Efficiency analysis with Real-World Data, GECCO

---

### Meta-Review · Area_Chair_Q8hw · 2024-02-05

**Recommendation:** Accept (Poster)
**Confidence:** 5

**Metareview:**

This paper describes a successive rank-based task assignment algorithm for online multiagent planning named SKATE. SKATE extends a previously developed metaheuristic approach to online task assignment called MinPos to allow multiple tasks to be assigned to a given agent at a given time step by incorporating and successively applying a task-to-agent ranking scheme. Theoretical complexity bounds of the algorithm are presented and an experimental comparative analysis to 2 existing online planning approaches, an ILP and a GA-based approach, is carried out. SKATE is shown to outperform these approaches in high load circumstances and exhibit better scaling properties to large sets of agents. An extended variant the utilizes variable receding horizons to expand the search is shown to further improve SKATE’s performance.

Strengths: The paper is well-written and is technically sound. The online planning problem addressed is well-motivated. The experimental analysis is likewise well formulated, and the results are compelling.

Weaknesses: First, the work is very incremental, amounting to a simple extension to the pre-existing MinPos algorithm to successively allocate multiple tasks to agents in situations where the input task load is high, and the degree of novelty and innovation is questionable. Second, a few important points related to reproducibility (e.g., how the parameter alpha is set, what is the rank matrix in Alg. 2), and prior related work (e.g., relationships to online VRP and TSP solution methods) are not discussed.

The reviewer questions and subsequent author(s) rebuttal identify several points of discussion that should be added to improve the paper. Please take care to incorporate these changes in the final version if the paper is accepted.

**Ethical Considerations:**

(1) Not Applicable: The paper does not have any ethical considerations to address